# Strategies for Biocontrol of *Listeria monocytogenes* Using Lactic Acid Bacteria and Their Metabolites in Ready-to-Eat Meat- and Dairy-Ripened Products

**DOI:** 10.3390/foods11040542

**Published:** 2022-02-14

**Authors:** Irene Martín, Alicia Rodríguez, Josué Delgado, Juan J. Córdoba

**Affiliations:** Higiene y Seguridad Alimentaria, Facultad de Veterinaria, Instituto Universitario de Investigación de la Carne y Productos Cárnicos, Universidad de Extremadura, Avda de las Ciencias, s/n, 10003 Cáceres, Spain; iremartint@unex.es (I.M.); aliciarj@unex.es (A.R.); jdperon@unex.es (J.D.)

**Keywords:** *L. monocytogenes*, LAB, bacteriocins, protective cultures

## Abstract

*Listeria monocytogenes* is one of the most important foodborne pathogens. This microorganism is a serious concern in the ready-to-eat (RTE) meat and dairy-ripened products industries. The use of lactic acid bacteria (LAB)-producing anti-*L. monocytogenes* peptides (bacteriocins) and/or lactic acid and/or other antimicrobial system could be a promising tool to control this pathogen in RTE meat and dairy products. This review provides an up to date about the strategies of use of LAB and their metabolites in RTE meat products and dairy foods by selecting the most appropriate strains, by analysing the mechanism by which they inhibit *L. monocytogenes* and methods of effective application of LAB, and their metabolites in these kinds of products to control this pathogen throughout the processing and storage. The selection of LAB with anti-*L. monocytogenes* activity allows to dispose of effective strains in meat and dairy-ripened products, achieving reductions form 2–5 logarithmic cycles of this pathogen throughout the ripening process. The combination of selected LAB strains with antimicrobial compounds, such as acid/sodium lactate and other strategies, as the active packaging could be the next future innovation for eliminating risk of *L. monocytogenes* in meat and dairy-ripened products.

## 1. Introduction

*Listeria monocytogenes* is one of the most important pathogenic microorganisms and is responsible for causing listeriosis, an illness that affects mainly pregnant women, new-borns, the elderly, and individuals with compromised immune systems [1]. Although it is a relatively rare illness, with a notification rate of 0.46 cases per 100,000 people in 2019 in the European Union (EU), most of the infections required hospitalisation (92.1%) [2]. This microorganism is a serious concern in the ready-to-eat (RTE) meat and dairy products industries, including dry-cured fermented sausages or ripened cheeses [3,4,5], since it could colonize and grow in raw material and pre-processed products throughout the processing and/or storage of these products, posing a risk for the consumers and/or also provoking non-compliance of microbiological criteria for this pathogen bacterium. Although in most of these RTE ripened foods, the reduction of water activity (a_w_) and pH throughout the ripening are hurdles that aid to control *L. monocytogenes*, this pathogen has been involved in many outbreaks linked to the consumption of the above products [3,6,7,8].

The use of lactic acid bacteria (LAB) as protective cultures could be an additional tool to control *L. monocytogenes* in RTE meat and dairy-ripened products. LAB have been frequently used as starter or protective cultures due to their natural ability to dominate the microbial population of many foods where they naturally occur due to their ability to catabolize carbohydrates to lactic acid and produce other biologically active compounds, such as organic acids, diacetyl, hydrogen peroxide, and antibacterial peptides and flavour precursors [9]. In addition, screening natural LAB strains to find the ones able to produce antimicrobial molecules is a promising strategy. An important number of either bacteriostatic or bactericidal compounds produced by LAB has been described [10].

The LAB genera are *Carnobacterium*, *Lactococcus*, *Leuconostoc*, *Oenococcus*, *Pediococcus*, *Streptococcus*, and the former *Lactobacillus* genus, which has been recently reclassified into 25 new genera [11,12]. Most of them have the status Generally Recognised as Safe (GRAS) according to the U.S. Food and Drug Administration (FDA). In addition, many LAB species have the recognition of Qualified Presumption of Safety (QPS) from the European Food Safety Authority (EFSA) (Table 1); thus, they have this presumptive qualification of being safe to be used as protective cultures in foods.

Although many LAB strains have been isolated and selected for their ability to in vitro inhibit *L. monocytogenes*, not all of them have been effectives in real RTE ripened food systems. In the present work, the strategies of selection of effective LAB species against *L. monocytogenes* to be used in RTE meat products and dairy-ripened products will be reviewed. In addition, we will also review the mechanism by which they inhibit *L. monocytogenes* as well as effective methods of LAB application and their metabolites in these kinds of products to control this pathogen throughout their processing and storage.

## 2. Selection and Evaluation of LAB from RTE Meat and Dairy-Ripened Products with Anti-*L. monocytogenes* Activity

Many traditional RTE fermented foods constitute rich ecological niches for screening LAB with anti-*L. monocytogenes* activity [13,14,15]. The selection of LAB with anti-*L. monocytogenes* activity from these products to be used as protective cultures should be performed from strains adapted to the ecological niche of these products since they must survive and are competitive in conditions of processing and/or storage. Thus, LAB strains should be isolated during ripening and/or storage conditions of meat and dairy-ripened products following different steps (Figure 1). First, LAB isolate should be obtained from meat and dairy-ripened products, testing different days of ripening (for example initial, half, and final time) or in different days of storage, with the purpose to obtain strains adapted to characteristics and processing conditions of the chosen foods. Then, isolated strains should be preliminary characterized by Gram staining, catalase reaction, shape by microscopic observation of overnight cultures, and biochemical features, as it has been reported in LAB strains isolation from soft cheese [16,17] or dry-cured fermented sausages [18]. After the first preliminary characterization, the isolates are screened for anti-*L. monocytogenes* in culture media usually by the agar spot-on-a-lawn method [14,19]. Thus, preliminary active isolates against *L. monocytogenes* are obtained, which are characterized by 16S rRNA sequencing [17] and further evaluated by co-inoculation with *L. monocytogenes* for anti-listerial activity in food models simulating temperature, water activity, and pH conditions of RTE products, as it has been reported by Martín et al. (Unpublished data) in soft cheese model for the selection of active anti-listerial LAB strains (Figure 1). This step is of the utmost importance to discard LAB strains lacking activity or with low activity in the processing or storage conditions of RTE foods. Thus, active LAB strains against *L. monocytogenes* able to be finally evaluated in RTE products are selected. In parallel, the selected LAB strains should be finally characterized by some additional method to the 16 rRNA sequencing, which allow the differentiation at strain level. This should be of great value to evaluate implantation of selected strains in RTE foods in the next step. Pulsed-field gel electrophoresis analysis (PFGE) has been reported as an appropriate method for the differentiation of LAB strains [20]. In addition, the RFLP analysis of the *tuf* gene has also been described as a suitable tool for the differentiation of LAB strains [21].

Finally, it should be investigated the ability of the most effective LAB strains to control the growth of *L. monocytogenes* in the food product by the challenge test methodology (Figure 1), as it has been reported in dry-cured fermented sausages [22] and ripened cheeses [16]. These authors have found the ripening reduction of *L. monocytogenes* from 2 to 5 log CFU/g because of the action of selected LAB strains. In addition, the assayed strains were successfully implanted in the food matrices.

Many studies have focused on the selection of LAB to reduce or eliminate *L. monocytogenes* in foods [23]. Campagnollo et al. [16] reported the isolation and characterization of LAB strains with anti-listerial activity and their effects on *L. monocytogenes* during refrigerated shelf-life of soft and ripening of semi-hard cheese. De Carvalho et al. [24] tested several LAB isolated from naturally fermented Italian salami for antagonistic activity against *L. monocytogenes*. Pedonese et al. [25] stated that *Lactilactobacillus* (*Ll*) *sakei* is capable of suppressing the growth of pathogenic and spoilage microorganisms and improving the sensory quality of fresh meat preparations and products.

Most of the selected LAB strains with anti-*L. monocytogenes* activity have also been evaluated for possible modification of physicochemical (pH, humidity content, a_w_), biochemical (proteolysis, lipolysis, volatile compounds generation), and sensorial parameters before being proposed as protective or starter cultures [16,26]. This evaluation allows to rule out some LAB strains that could negatively affect the physicochemical properties of RTE products and propose only those strains without effect or with positive repercussion on sensorial characteristic. All the former works have selected and characterized LAB strains that are available to be used as protective cultures in RTE meat and dairy-ripened products due to their effect against *L. monocytogenes*.

## 3. Effect of Selected LAB Strains on *L. monocytogenes* Inhibition

LAB mechanisms for *L. monocytogenes* inhibition in RTE foods include (a) production of inhibitory compounds, (b) competition for nutrients, (c) prevention of pathogen adhesion, and (d) competition for space or niche competition.

### 3.1. Production of Inhibitory Compounds

LAB have the ability to produce antimicrobial compounds, such as lactic acid and other organic acids, ethanol, diacetyl, carbon dioxide, hydrogen peroxide, bacteriocins, or bactericidal proteins [27]. Table 2 shows a summary of these compounds and their mechanisms of action.

Homofermentative LAB ferment carbohydrates to produce lactic acid as the major metabolic product, leading to pH reduction of food and also directly to growth inhibition of many microorganisms [32]. It has been described that the principal antimicrobial compound responsible for their activity against pathogens is synthesis of organic acids, mainly lactic and acetic acids [28,33]. Organic acids act by acidifying the intracellular pH, generating an unfavourable local microenvironment for pathogenic bacteria [28,34]. They also act by inhibiting the active transport of excess internal protons that leads to the depletion of cellular energy [35]. The bacterial cell wall and the cytoplasmic membrane are the main targets of organic acids provoking alteration and death and metabolic functions of pathogenic microorganisms [36]. It has been proven that concentrations of 0.5% (*v*/*v*) of lactic acid could completely disrupt the growth of pathogenic microorganisms, such as *Salmonella* spp., *Escherichia coli*, or *L. monocytogenes* [37]. Wemmenhove et al. [38] tested the effect of lactic acid against *L. monocytogenes* in Gouda cheese. In addition, it has been studied that the short-chain fatty acids produced by LAB in food fermentation improve the integrity of the barrier and prevent the adhesion of pathogenic bacteria or indirectly inhibit the expression of the virulence genes at the transcriptional level of *L. monocytogenes*.

However, heterofermentative LAB produce lactic acid and additional compounds, such ethanol and carbon dioxide [39,40]. Ethanol produced by heterofermentative LAB affects membrane fluidity and integrity, leading to plasma membrane leakage and causing bacterial death [28]. Barker and Park [41] found that a 5% ethanol concentration inhibited the replication of *L. monocytogenes*.

Some LAB strains produce hydrogen peroxide (Table 2) that provokes inactivation of essential biomolecules of the pathogens, such as *L. monocytogenes*, by superoxide anion chain reaction negatively affects its viability [29]. Thus, in vitro inhibition of *L. monocytogenes* mainly due to production of hydrogen peroxide was reported by Ghalfi et al. [42] with an *Ll.*
*curvatus* selected strain from meat origin.

Heterofermentative LAB strains also produce carbon dioxide as a by-product of sugar fermentation that inhibits growth of *L. monocytogenes* [27,30].

Some LAB strains produce diacetyl, which interferes with arginine utilization of pathogen microorganisms, such as *L. monocytogenes* [29]. In fact, the combination of some bacteriocins, such as reuterin and diacetyl, have been reported to be anti-microbial additives with effects against *L. monocytogenes* [43].

Bacteriocins are produced by some LAB strains and contribute to the biological control of pathogenic and spoilage microorganisms. Bacteriocins and their effect against *L. monocytogenes* are detailed in the following section (Section 4).

### 3.2. Competition for Nutrients

One of the main mechanisms of action of non-pathogenic bacteria against pathogenic bacteria is competition for nutrients in a specific niche, leading to depletion [44]. The metabolic activity of *L. monocytogenes* may not be affected by antimicrobial compounds produced by LAB (bacteriocins, organic acids including lactic, and acetic acids) due to its acid tolerance and synthesis of proteolytic enzymes. Therefore, the growth rate of LAB is of great importance in their role in competing for nutrients with *L. monocytogenes* [45].

However, under stress conditions provoked by the lack of nutrients or acid stress caused by the organic acids synthesized by LAB, *L. monocytogenes* can express some of its virulence factors, such as InlA and InlB protein, to confront this stress [46]. In the same way, LAB may induce the synthesis of bacteriocins as a method to minimize the stress caused by insufficient nutrients [44].

### 3.3. Competition for Space

Another mechanism of action of LAB includes the competitive exclusion of pathogenic microorganisms from space [47]. Adhesion of *L. monocytogenes* on host cells is of great importance for their invasion and virulence [44,48]. LAB can prevent the binding of *L. monocytogenes* on host cells by colonizing the host cells and/or saturating the pathogen binding receptor [44]. Corr et al. [49] showed that pre-treatment of intestinal epithelial cells with LAB before infection with *L. monocytogenes* resulted in a significant decrease in its invasion (60–90%). When there is a direct cell-to-cell competition between *L. monocytogenes* and LAB to the binding sites, LAB inhibit the attachment of the pathogen, being reduced by 4.38 and 3.22 log CFU/g after 24 h and 72 h, respectively [50]. According to Pilchová et al. [51], a significant inhibition of the adhesion, invasion, and transepithelial translocation of *L. monocytogenes* was obtained using *Lacticaseibacillus (Lc) paracasei* but only if this strain was recombined to obtain the expression of the adhesion protein of *L. monocytogenes.*

Competitive inhibition of selected LAB strains has been reported in biofilm formation of *L. monocytogenes*, which poses a risk factor in the food industry [52]. These authors found that selected *Ll. curvatus*, *Lactococcus (La) lactis*, *Lactobacillus helveticus*, and *Weissella viridescens* isolated from Brazilian’s foods developed protective biofilms against *L. monocytogenes* hampering the biofilm formation by this pathogen, mostly due the exopolysaccharide production by these LAB strains. Thus, selected LAB strains could be promissory candidates for controlling the presence of *L. monocytogenes* biofilms in food-processing facilities [50].

### 3.4. Reduction of L. monocytogenes Virulence by LAB

Many authors have studied the use of LAB to reduce the expression of virulence of pathogens by modulating the expression of genes or proteins through bacterial signalling mechanisms. Thus, Dutra et al. [53] reported that *Lc. casei* and *Lc.*
*rhamnosus* significantly reduced the binding (10–13%) and invasion (40–50%) of *L. monocytogenes* into cells, indicating that LAB are effective in reducing this pathogen colonization both when administered prophylactically and during infection. Another study by Upadhyay et al. [54] demonstrated that *Liminosilactobacillus (Li) reuteri*, *Li. fermentum*, *Lactiplantibacillus (Lp) plantarum*, and *La. lactic* reduced the adhesion and invasion of Caco-2 cell of *L. monocytogenes*, down-regulating the expression of the majority of virulence genes of this pathogen (*plcA*, *plcB*, *iap*, *hly*, *inlA*, *inlB*, *actA*, and *prfA*).

A significant reduction in *L. monocytogenes* virulence on epithelial cells was observed when the cell monolayers were mixed with *Carnobacterium divergens* V41 cultures during 1 or 4 h. The ability to control foodborne pathogenic microorganism virulence has previously been evaluated for probiotic LAB and found to be strain specific. For example, Garriga et al. [55] reported a that bacteriocinogenic *Ll. sakei* strain significantly decreased the adhesion of *L. monocytogenes*.

Winkelströter and De Martinis [56] showed that the bacteriocins produced by *Enterococcus* (E) *faecium*, *Leuconostoc (Le) mesenteroides*, and *Ll. sakei* significantly decreased the expression of *inlA* gene from different *L. monocytogenes* strains.

## 4. Bacteriocins with Activity against *L. monocytogenes*

Antimicrobial peptides or proteins produced by LAB are small, ribosomally synthesized, and possess activity against closely related Gram-positive bacteria, whereas producer bacteria are immune to their own proteinaceous metabolites [57]. In general, bacteriocin-producing strains mostly belong to the formerly named *Lactobacillus* and *Lactococcus* genera and are well-proven to have Generally Recognised as Safe (GRAS) status [58].

The antibacterial spectrum of bacteriocins frequently includes spoilage microorganisms and foodborne pathogens, such as *L. monocytogenes* and *Staphylococcus aureus*. In addition to their antimicrobial action towards these unwanted species, bacteriocins are believed to contribute to increasing the competitiveness of the producer strain [59].

So far, bacteriocins have been classified into four general classes attending to their composition and structural properties. The first class, termed as lantibiotics, contains unusual amino acids (i.e., lanthionines and β-methyllanthionines). The production of this class of bacteriocins involves post-translational modifications, which are well described for nisin [60]. The second one is comprehended by bacteriocins that do not contain lanthionine residues, being characterised by heat stability and their site of action as the cell membrane. The third one is composed by the large and heat-sensitive bacteriocins, and finally, the fourth class is bacteriocins containing other chemical moieties (carbohydrates and lipids) [61]. Subsequent subclassifications into these groups has been performed, as shown in Table 3.

Among all bacteriocins, the post-translationally modified class Ia nisin is probably the best-known bacteriocin with listericidal effect. Nisin’s mechanism of action involves membrane permeabilization through binding to lipid II, the phenyl chain-linked donor of the peptidoglycan building blocks [80]. This lipid II is believed crucial to peptidoglycan synthesis, and nisin is considerably more active towards peptidoglycan-rich, Gram-positive microorganisms than Gram-negative ones, the latter being only affected by nisin in conjunction with chemically induced damage of the outer membrane [81]. This bacteriocin is approved as a preservative by the European Commission, named as E 234, intended for use in various dairy products, among others [82].

Apart from the well-known nisin, the II class bacteriocins and most concretely IIa class are the most commonly active against *L. monocytogenes*. Some of these bacteriocins include garviecin LG34, bifidocin A, leucocin C-607, pediocin GS4, plantaricin LPL-1, or pediocin PA-1 or sakacins [65,83,84,85,86,87]. These pediocin-like class IIa bacteriocins deploy great bacterial inhibition at nanomolar concentrations in relation to the high affinity to specific receptors or docking molecules [83]. One of these target receptors is the called mannose-phosphotransferase system (man-PTS), which phosphorylates and transports carbohydrates and other related substances, and the membrane components, ManY/IIC and ManZ/IID, belonging to man-PTS, form a membrane-located complex [88,89,90]. Additionally, these class IIa bacteriocins act on the cytoplasmic membrane of Gram-positive microorganisms, dissipating the transmembrane electrical potential and resulting in intracellular ATP depletion. Furthermore, they induce the leak of ions, amino acids, proteins, and nucleic acids by forming hydrophilic pores in target membranes [87,91].

Although the nature of these compounds is able to inhibit *L. monocytogenes*, several strains from this pathogen have been able to develop a certain degree of resistance against bacteriocins. The two strategies deployed by resistant bacteria to counteract the bacteriocins effect are membrane surface charge and membrane fluidity [92,93].

For class I bacteriocins, such as nisin, changes in membrane lipid composition are involved in *L. monocytogenes* resistance [94], as well as phospholipids charges in interactions between artificial membranes and nisin [95,96]. For class IIa bacteriocins, their target, the man-PTS receptor, plays a key role in the resistance against this type of bacteriocins [97]. On one hand, the low expression of genes related to this receptor is directly linked to IIa-class bacteriocin resistance [98]. On the other hand, as occurs for leucocin A, changes in membrane fatty acid composition, increase in D-alanine content of wall teichoic acid, and increase in L-lysine content of membrane phospholipids are other common strategies elicited by class-IIa resistant *L. monocytogenes*.

To overcome the limitation of bacteriocin resistance by *L. monocytogenes* and maximize its inhibitory activity, the use of different bacteriocins combined or even a given bacteriocin in combination with technological or chemical treatments could provide an alternate approach to tackle this problem [99], enhancing the antimicrobial effect as discussed in Section 6.

### Selection of Bacteriocin-Producing Lactic Acid Bacteria and Bacteriocin Characterization

A common requirement for any protective culture is the safety of these organisms irrespectively of the production of antimicrobial metabolites. The ability to produce biogenic amines, such as 2-phenylethylamine, putrescine, cadaverine, agmatine, spermine, spermidine, histamine, and tyramine, should also be ruled out by gene analyses [100]. Additionally, other unwanted genes involved in virulence, such as *asa1*, *agg*, *efaA*, *hyl*, *esp*, *cylL_L_*, *cylL_S_, ace*, and *gelE*, should also be tested with the aim to ensure the lack of virulence in the selected strain [101]. Finally, the antibiotic resistance must be evaluated both for the possible involvement of some of these bacteria, mainly *E. faecalis* and *E. faecium*, in human infections [102] and the induction of potential antimicrobial resistance through horizontal gene transference [103].

Once the safety characterisation has been considered, the first step to evaluate the production of bacteriocins would entail a screening based on the assessment of the antimicrobial activity of the cell-free medium (CFM) in which the candidate LAB has been grown. The most recommended conditions to maximize the bacteriocin in vitro production for LAB are Man Rogosa Sharpe broth, pH 5.5–6.5, at 30–37 °C for 24–48 h [104]. After obtaining the CFM, the in vitro antimicrobial activity against *L. monocytogenes* should be tested by co-culturing, in which CFM is simultaneously added with *L. monocytogenes*, or by delayed culturing, in which broth medium is inoculated with the pathogen and incubated, followed by CFM addition after 6 h of incubation [105].

A relatively quick and cheap technique for bacteriocin characterization for those CFM showing any degree of inhibition on *L. monocytogenes* is the tricine SDS-PAGE analysis [106] after protein precipitation with 40, 60, and 80% ammonium sulphate [107]. This tool is useful for a primary characterization given that it informs about the presence or absence of any proteinaceous compound as well as displays information about the potential bacteriocin molecular weight if any band is found. Thus, it serves to categorize the potential bacteriocin within some of the compatible classes attending to this feature. Although it could be thought that the main limitation of this technique is the degree of purity of the proteinaceous precipitation, since generally a complex of proteinaceous compounds is excreted to the CFM, a simple method based on the evaluation of the anti-listerial activity of every band from a given sample by setting the tricine-SDS-PAGE onto a solid medium inoculated with *L. monocytogenes* is commonly used [108].

Whether the tricine-SDS-PAGE reveals a single band linked to a sample with anti-listerial activity, apart from the information about the molecular weight, this band could be excised, digested, and analysed by high-resolution mass spectrometry to identify its aminoacidic sequence and similarities with other previously published bacteriocins by means of software analyses such as MASCOT [109].

With the aim to genetically characterize the ability of bacteriocin production by LAB, the detection of genes that encode for these metabolites deserves to be exploited. There are numerous target genes to evaluate its presence in food products and even the bacteriocin production through transcriptional analysis. Some of these genes have been used for characterizing LAB ability to produce bacteriocins [110,111]. However, to completely characterize the ability of bacteriocin production, the most recommended approach consists of whole-genome sequencing to evaluate the presence of any reported bacteriocin [107]. This tool additionally offers the possibility of sweeping the currently sequenced genome in the future with the aim to detect genes encoding for ulteriorly discovered bacteriocins.

The bacteriocins as bioprotective tool against *L. monocytogenes* could be split into two different applications: (a) the addition of the purified bacteriocin to the food and (b) the inoculation of the bacteriocin-producing LAB in the food. The former has been assayed with the well-known nisin to successfully inhibit *L. monocytogenes* in milk although the anti-listerial effect depends on its chemical composition and the technological process at which the food has undergone [112]. Additionally, intrinsic mechanisms from milk to inhibit pathogens, such as lactoperoxidase, seem to work synergistically with this bacteriocin, resulting in maximizing the anti-listerial effect [113].

## 5. Application of Selected LAB or Bacteriocins in RTE Dry-Cured Meat Products

The meat industry has carried out extraordinary research efforts to minimize the appearance of outbreaks caused by foodborne *L. monocytogenes*. The application of selected LAB and/or their purified antimicrobial metabolites for the biopreservation of RTE dry-cured meat products has been increasing in the last years with promising results. Selected LAB strains or their metabolites have been directly incorporated into the meat products throughout the processing to reduce the hazard posed by the presence and growth of *L. monocytogenes* in these products.

With this aim, *Ll. sakei* has been widely employed in several studies with different results. García-Diez and Patarata [114] concluded that the addition of *Ll. sakei* at a concentration of 6 log CFU/g did not provoke significant reduction in *L. monocytogenes* counts in a Portuguese dry-fermented sausage. However, Ortiz et al. [115] showed that *Ll. sakei*, when added to meat batter in Iberian chorizo, showed an anti-listerial activity at either 7 or 20 °C, reducing by 2 log_10_ units the pathogen counts. In addition, Vaz-Velho et al. [116] demonstrated that *Ll. sakei* was enough to minimise *L. monocytogenes* counts (up to 2 log CFU/g) in a Portuguese salami-like product, Alheira. Selected *Lp. plantarum* has also been used to inhibit and control *L. monocytogenes* in RTE meat products. Thus, Kamiloglu et al. [117] evaluated the effect of five *Lp. plantarum* (initially inoculated at 7 log CFU/g) against *L. monocytogenes* in sucuk, a traditional dry-fermented sausage from Turkey. They observed a decrease in *L. monocytogenes* counts from 1 to 2.7 log CFU/g for the different *Lp. plantarum* strains tested during ripening. In such work, they determined that acidification and production of bacteriocins and/or bacteriocin like peptides were the cause for the control of this pathogenic microorganism. Zanette et al. [118] tested the anti-listerial activity of two *Lp. plantarum* strains (one bacteriocin-producing strain and one bacteriocin non-producing strain) and found they were equally effective to limit *L. monocytogenes* growth (≈1.7 log CFU/g reduction) from the initial levels of the pathogen (4 log CFU/g).

The combination of selected active LAB, such as *Ll. sakei* (CRL1862), with bacteriocin combination and 2.5% lactic acid and acetic acid diminished the *L. monocytogenes* counts at levels lower than 2 log CFU/g (from initial counts at 3–4 log CFU/g) in frankfurters from day 6 to day 36 at 10 °C [119]. However, no significant additional reductions were observed when selected *Ll. sakei* was evaluated in combination of packing under vacuum or modified atmosphere packaging. Nikodinoska et al. [120] tested the antagonistic activity of *Lp. plantarum* alone and combined with nitrite (at two concentrations) against the pathogenic bacterium in a chorizo sausage model. Counts of *L. monocytogenes* were reduced with the addition of the LAB strain (ranging from 2.6 to 3.8 log CFU/g depending on the nitrite concentration used). In samples where nitrite was not added, *Lp. plantarum* reduced *L. monocytogenes* growth but not until the end of ripening. On the contrary, Macieira et al. [121], who used bacteriocinogenic *Lp. plantarum* cultures (at a concentration of 6 log CFU /g) in a traditional Portuguese fermented dry-cured sausage, did not have any antagonistic activity against *L. monocytogenes* (initially inoculated at 5 log CFU/g).

In the study carried out by Sadaghiani et al. [122], they checked the effect of one strain of *Lp. plantarum* (initially inoculated at 7 log CFU/g) in ground raw beef alone and in combination with a garlic extract (1%). The LAB strain alone decreased the counts of the pathogen at 0.7 log CFU/g, but when combined with the garlic extract, this reduction was 1.5 log CFU/g.

*Pediococcus (P) acidilactici* has also been quite utilised as a biopreservative to control the development of *L. monocytogenes* in RTE meat products. Cosansu et al. [123] demonstrated that the bacteriocin-producing *P. acidilactici* possessed a significant anti-listerial activity on sucuk but not on sliced turkey bread. *P. acidilactici* produced a reduction of 3.3 log CFU/g *L. monocytogenes* counts after 8 days of sucuk fermentation at mild temperatures (22–24 °C). On the other hand, Ortiz et al. [115] showed that a starter culture containing *P. acidilactici* in Iberian chorizo provoked an anti-listerial effect at 7 °C.

Other researchers have focused on looking for other LAB species as biopreservatives to counteract and minimize the growth of *L. monocytogenes* in RTE meat products. Regarding *P. pentosaceus*, it was added individually and in combination with *P. acidilacti* in sliced fresh beef samples [124]. This study concluded that the use of *P. pentosaceus* alone or combined with *P. acidilacti* is promising since they limited the *L. monocytogenes* counts <2 log CFU/g on day 2. *Li. reuteri* is another LAB species used as biopreservative in the meat industry. Sadaghiani et al. [122] checked the anti-*L. monocytogenes* activity of a *Li. reuteri* strain in conjunction with garlic extract (1%) in beef, concluding that the combination of garlic extract with *Li. reuteri* caused a 1.4 log count reduction, while *Li. reuteri* alone only provoked a 0.5 log reduction. Orihuel et al. [125] reported that a bacteriocinogenic *E. mundtii* strain had limited anti-*L. monocytogenes* activity in beef sausage when applied alone, but in combination with curing additives, reductions of 2 log CFU/g counts were achieved. Finally, Castellano et al. [119] showed that the bacteriocin synthesized by *Ll. curvatus* possessed some bacteriostatic effect in frankfurters but lower than that shown by the bacteriocin produced by *Ll. sakei.*

Some metabolites synthesized by LAB have also been utilised as a biopreservative to control *L. monocytogenes* in RTE meat products. Trinetta et al. [126] studied the antagonistic effect of sakacin A, a bacteriocin produced by *Aureobasidium pullulans*, when it was directly added to RTE turkey breasts and when incorporated in a pullulan film to package this product. Results showed that sakacin A directly applied to turkey decreased the *L. monocytogenes* counts by more than 2 log CFU/g, while sakacin A-containing pullulan films diminished its counts 3 log CFU/g. Another bacteriocin that displayed anti*-L. monocytogenes* activity was nisin when was added in RTE turkey ham [127]. This bacteriocin was used in different concentrations (from 0.2 to 0.5%), and its antagonistic effect increased as the concentration did, keeping the *L. monocytogenes* counts lower than the control in all treatments. Leucocin A is another bacteriocin used for *L. monocytogenes* control purposes in RTE meat products. This bacteriocin produced by *Le. gelidum* has been employed in wieners (sausages) to counteract *L. monocytogenes* [128]. The antimicrobial activity of this bacteriocin was lower than the previous ones, obtaining only a reduction of 1 log CFU/g after 16 days of incubation at refrigeration temperatures.

## 6. Application of Selected LAB or Bacteriocins in RTE Dairy-Ripened Products

Most of the application of LAB species in dairy-ripened products have been reported in cheese throughout the ripening or storage. Thus, selected strains of *Ll. sakei* and *Lp. plantarum* used as protective cultures in soft cheese reduced the loads of *L. monocytogenes* from 0.5 to almost 1 log CFU/g during 1375 h of storage at 15 °C [17]. Higher reduction was found in semi-hard cheeses ripened with *L. brevis*, *Lp. plantarum*, and *E. faecalis*, where *L. monocytogenes* counts were reduced by 4 log CFU/g after 15 days of ripening in cheeses made with raw milk and after 21 days in cheese made with pasteurized milk [16].

Selected *Lactococcus* spp. has been widely used as protective cultures in cheese. Thus, Kondrotiene et al. [129] found a significant reduction in *L. monocytogenes* counts when three nisin A-producing *La. lactis* strains were applied to fresh cheese. In addition, selected strains of *La. lactis* subsp. *lactis* and *E. durans* as individual or mixed cultures have also been reported to provoke a reduction of 2–3 log CFU/g of *L. monocytogenes* during 35 days of storage at 4 °C of ultrafiltered cheese [130]. These authors underlined the potential application of the above LAB strains in bio-control of this pathogen bacterium during storage of ultrafiltered cheese.

*Ll. sakei*, *La. lactis*, and *Carnobacterium* strains selected from Gorgonzola cheese have been reported to provoke a notable inhibition at low level of contamination of *L. monocytogenes* (2 log CFU/g) in this kind of cheese [10]. This inhibition was found during the first stage of ripening (6 days), and *L. monocytogenes* cells were maintained below the EC limit (<2 log CFU/g) for 60 days. However, these authors reported that when *L. monocytogenes* was inoculated on the cheese surface at the end of ripening process (after 50 days; pH: 6.7), only one of the selected *La. lactis* strains exerted a significant inhibition on the growth of this pathogen if the cheese was strictly maintained at 4 °C.

Morandi et al. [10] underlined that the susceptibility of *L. monocytogenes* biotypes to LAB antimicrobial activity is strain dependent. Thus, a blend of different LAB strains could represent a more effective tool to develop protective culture for ripened cheeses. In this sense, combinations of different LAB strains have been proposed to be used as protective cultures in cheese. The combination of *Lp. plantarum* strain (initially inoculated at 8 log CFU/mL) with a nisin producer reduced *L. monocytogenes* to undetectable levels in cheese by day 28 of ripening [131]. Furthermore, these authors found that *Lp. plantarum* was much more effective in inhibiting *L. monocytogenes* when the nisin producer was attached than when it was alone.

Some studies have reported the use of bacteriocin produced by LAB for biopreservation of cheeses [132,133]. Nisin is the most frequently used although it has been reported as efficient in control *L. monocytogenes* only in fresh cheese [134,135]. An increase in anti-*L. monocytogenes* activity has been suggested when combining nisin with a second bacteriocin [134]. Therefore, the use of nisin in combination with the IIa class bovicin HC5 in fresh cheese against *L. monocytogenes* has been reported to provoke a 4 log reduction of this pathogen after 9 days at refrigeration storage [136]. In ripened cheese, it has been proposed as most effective to use nisin-producing strain of *Lc. lactis* subsp. lactis for the milk before cheese production, provoking an initial reduction higher than 2 log CFU/g [137] since the use of nisin could have the problem of the regrowth during ripening of the surviving *L. monocytogenes* [134]. Other bacteriocins, such as pediocins, enterocins, and lactacins, have also been used on the surface of cheese and mainly in fresh cheese [138,139,140], but their utility in ripened cheese is limited. Thus, although it has been highlighted that the utilization of bacteriocins could contribute to the creation of low-salt and healthier formulations of cheeses and to the optimization of processing conditions without compromising the microbiological safety of these RTE foods [141], the problem of the regrowth during ripening of surviving *L. monocytogenes* should be considered, which it makes more effective the use of selected LAB than the direct addition of bacteriocins.

Furthermore, combinations of different preservation methods may act synergistically or provide higher protection than a single method alone [142]. Thus, the combination of selected LAB with antimicrobial compounds has been proposed. In this sense, it has been proposed that selected *La. lactis* be used in combination with acid/sodium lactate (LASL-L-lactic acid 61% (*w/w*) and L-sodium lactate 21% (*w/w*)) [143]. The former authors found a total inhibition of *L. monocytogenes* strains in the first 50 days of ripening of Gorgonzola cheese when this combination was used, while LASL with selected *C. divergens* was more effective in the second part of ripening when the pH was raised. These authors encouraged the use of LASL along with antimicrobial LAB rotation schemes during cheese ripening for the prevention and/or control of the *L. monocytogenes* on the cheese surface of Gorgonzola cheese.

Finally, the use of active packaging with bacteriocins produced by selected LAB species is a promising strategy to control *L. monocytogenes* in packaged cheeses. In fact, Contessa et al. [144] described a film based on agar-agar incorporated with bacteriocin produced by a selected *Lc**. casei* to be used as active packaging in curd cheese. This active packaging provokes a reduction of 3 log_10_ units of pathogen bacteria, such as *L. monocytogenes*.

## 7. Conclusions and Future Remarks

*L. monocytogenes* is a serious concern in the RTE meat and dairy-ripened products industries. The use of LAB as protective cultures and/or their metabolites could be a promising tool to control *L. monocytogenes* in these kinds of products. Although LAB strains are present in most of the ripened foods as the natural microbial population, to find strains with anti-*L. monocytogenes* activity able to survive in conditions of ripened products, an appropriated selection methodology is necessary. This includes recovery of LAB isolates from different ripening/storage conditions and evaluation of the anti-listerial activity in food models simulating temperature, a_w_, and pH conditions of the processing. Then, final selection should be performed after evaluation of the most active strains in food matrices, following the challenge test methodology. As a result of the proposed isolation and selection methods for LAB strains with the ability to produce antimicrobial compounds, such as lactic acid and other organic acids, ethanol, diacetyl, carbon dioxide, hydrogen peroxide, bacteriocins, are available. In addition, the selected LAB strains can compete for nutrients and space with *L. monocytogenes* and some of them are able to eliminate this pathogen bacterium from biofilm and reduce its virulence and the ability of *L. monocytogenes* to survive. These strains have showed effectivity in meat and dairy-ripened products, achieving reductions form 2–5 log_10_ units of *L. monocytogenes* throughout the ripening process. This could be sufficient to guarantee the elimination of this pathogenic bacterium throughout the ripening/storage of RTE meat and dairy-ripened products when this pathogen contaminates these products at the usual levels (below 2 log CFU/g). This is of utmost importance since minimizing the risk of listeriosis caused by the consumption of these products improves food safety and meets the microbiological criteria of RTE foods throughout their shelf life. Bacteriocins could be also used to control *L. monocytogenes*, but their activity in these products could be limited by the regrowth during ripening or storage of the surviving strains of this pathogen. Thus, the combination of different active LAB strains and those bacteriocigenic ones could be the most appropriate strategies to control *L. monocytogenes* in ripened foods. Furthermore, the combination of selected LAB strains with antimicrobial compounds, such as acid/sodium lactate, and other strategies for active packaging could be the next step to eliminate the risk posed by *L. monocytogenes* in meat and dairy-ripened products.

## Figures and Tables

**Figure 1 foods-11-00542-f001:**
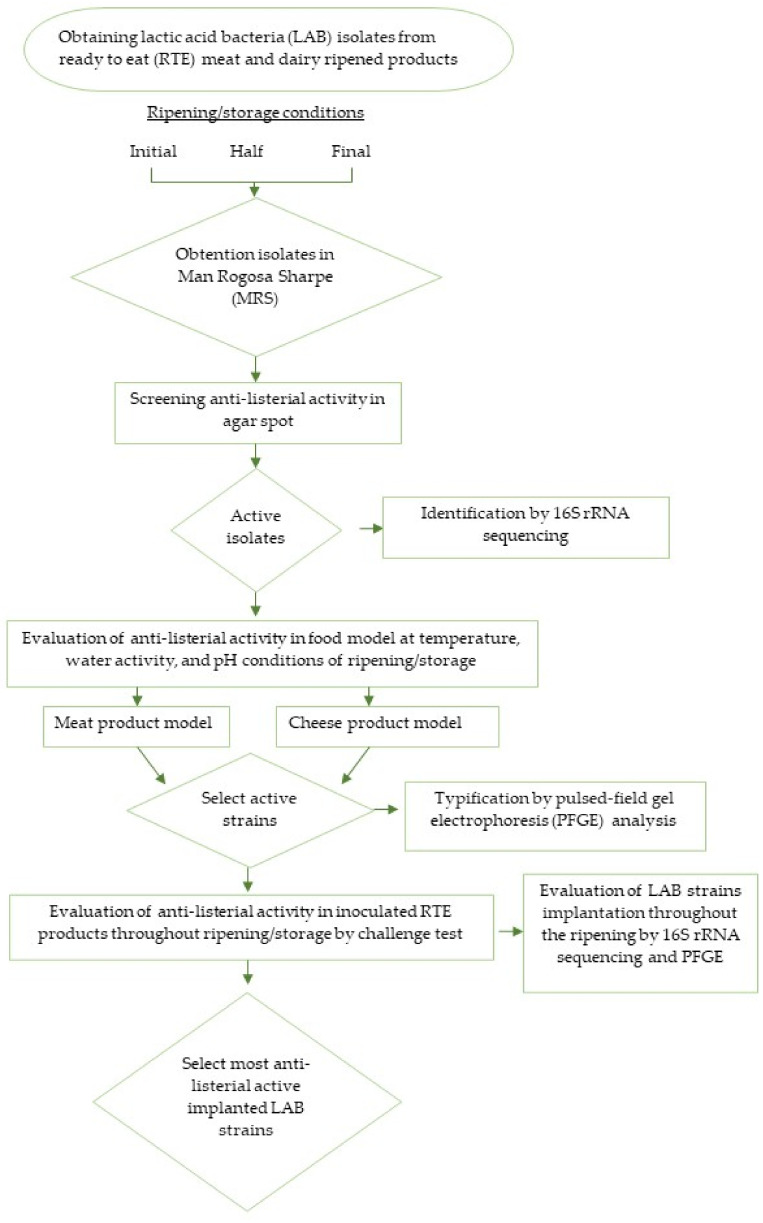
Diagrammatic flowchart for the isolation and selection of LAB strains with anti-listerial activity from RTE meat and dairy foods.

**Table 1 foods-11-00542-t001:** LAB included in the 2020 updated list of QPS status recommended biological agents for safety risk assessments carried out by EFSA Scientific Panels and Units [11].

*Bifidobacterium adolescentis*	*Lactobacillus delbruechkii*	*Ligilactobacillus animalis*
*Bifidobacterium animalis*	*Lactobacillus gallinarum*	*Ligilactobacillus aviaries*
*Bifidobacterium bifidum*	*Lactobacillus gasseri*	*Ligilactobacillus salivarius*
*Bifidobacterium breve*	*Lactobacillus helveticus*	*Liminosilactobacillus fermentum*
*Bifidobacterium longum*	*Lactobacillus johnsonii*	*Liminosilactobacillus mucosae*
*Carnobacterium divergens*	*Lactobacillus kefiranofaciens*	*Liminosilactobacillus panis*
*Companilactobacillus alimentarius*	*Lactococcus lactis*	*Liminosilactobacillus pontis*
*Companilactobacillus farciminis*	*Lapidilactobacillus dextrinicus*	*Liminosilactobacillus reuteri*
*Corynebacterium ammoniagenes*	*Latilactobacillus curvatus*	*Loigolactobacillus coryniformis*
*Corynebacterium glutamicum*	*Latilactobacillu sakei*	*Microbacterium imperial*
*Fructilactobacillus sanfranciscensis*	*Lentilactobacillus buchneri*	*Oenococcus oeni*
*Lacticaseibacillus casei*	*Lentilactobacillus diolivorans*	*Pasteuria nishizawae*
*Lacticaseibacillus paracasei*	*Lentilactobacillus hilgardii*	*Pediococcus acidilactici*
*Lacticaseibacillus rhamnosus*	*Lentilactobacillus kefiri*	*Pediococcus parvulus*
*Lactiplantibacillus pentosus*	*Lentilactobacillus parafarraginis*	*Pediococcus pentosaceus*
*Lactiplantibacillus plantarum*	*Lentilactobacillus paraplantarum*	*Propionibacterium acidipropionici*
*Lactobacillus acidophilus*	*Leuconostoc citreum*	*Propionibacterium freudenreichii*
*Lactobacillus amylolyticus*	*Leuconostoc lactis*	*Secundilactobacillus collinoides*
*Lactobacillus amylovorus*	*Leuconostoc mesenteroides*	*Streptococcus thermophilus*
*Lactobacillus cellobiosus*	*Leuconostoc pseudomesenteroides*	
*Lactobacillus crispatus*	*Levilactobacillus brevis*	

**Table 2 foods-11-00542-t002:** Inhibitory compounds produced by selected LABs and their mechanisms of action against pathogens microorganisms, such as *L. monocytogenes*.

Inhibitory Compound	Mechanism of Action	References
Lactic acid and other volatile acids	Disruption of cellular metabolism	[27]
Ethanol	Membrane fluidity and integrity	[28]
Hydrogen peroxide	Inactivation of essential biomolecules by superoxide anion chain reaction	[29]
Carbon dioxide	Anaerobic environment and/or inhibition of enzyme decarboxylation and/or disruption of the cell membrane	[30]
Diacetyl	Interference with arginine utilization	[29]
Bacteriocins	Disruption of cytoplasmic membrane	[27,31]

**Table 3 foods-11-00542-t003:** Bacteriocins classification, main features, examples of different bacteriocins, and their producer microorganisms.

Class	Characteristics	Example	Producer	Reference
Ia	Lantibiotics (<5 KDa)	Nisin	*Lactococcus lactis*	[62]
Ib	Carbacyclic lantibiotics	Labyrinthopeptien A1	*Actinomadura nambiensis*	[63]
Ic	Sactibiotics	Subtilosin A	*Bacillus subtilis*	[64]
IIa	Heat-stable peptides with N terminal- YGNGV	Pediocin PA-1, sakacins A and P, leucocin A, garviecin LG34	*Pediococcus pentosaceus*, *Pediococcus acidilactici*, *Lactilactobacillus sakei*, *Lactococcus garvieae*	[65,66,67,68,69]
IIb	Two-peptide bacteriocins	Lactococcin G, plantaricin EF and JK	*Lactiplantibacillus plantarum*, *Lactococcus* spp.	[70,71,72]
IIc	Circular bacteriocins	Enterocin AS-48, gassericin A	*Lactococcus gasseri*, *Enterococcus faecalis*	[73,74]
IId	Single, linear, nonpediocin-like bacteriocins	Thuricin S, bactofencin A	*Bacillus thuringensis*, *Ligilactobacillus salivarius*	[75,76]
IIIa	Heat labile, >30 KDa with hydrolase activity	Lysostaphin	*Staphylococcus. simulans* biovar *staphylolyticus*	[77]
IIIb	Heat labile, >30 KDa without hydrolase activity	Helveticin	*Lactobacillus helveticus*	[78]
IV	Large complexes with carbohydrate or lipid moieties	Enterocin F4-9	*Enterococcus faecalis*	[79]

## Data Availability

Not applicable.

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
