# Peer review of "Strategies for Biocontrol of Listeria monocytogenes Using Lactic Acid Bacteria and Their Metabolites in Ready-to-Eat Meat- and Dairy-Ripened Products"

_foods, 2022, doi:10.3390/foods11040542_

Round 1

Reviewer 1 Report

  • Kindly rephrase the manuscript as plagiarism is 30 % according to the Turnitin report.
  • The proposed article provides interesting results and adds to the knowledge base.
  • Although the overall manuscript is well written, there are some typographical and grammatical errors in the whole manuscript.
  • Kindly remove the references of submitted articles (Reference # 24 &30).
  • Kindly incorporate page numbers or article numbers of all the published articles.
  • Citations, references, layout, and format should be according to journal’s guidelines.

Author Response

We would like to thanks to the Reviewer 1 because of all the comments have been very helpful to improve quality of the paper. 

Reviewer 2 Report

This is an interesting review article that compiles studies on biocontrol of Listeria monocytogenes by use of LAB and their metabolites in meat and dairy products. This manuscript is worth reporting, but after some minor corrections. Please see my specific comments below.

Line 61: Please add a comma after “In the present work”

Fig.1: Abbreviations should be explained in the caption such as MRS, T, aw, PFGE. Although they are familiar to researchers, it is best to explain them.

Lines 160-162: Since the focus is L. monocytogenes, it should be explained that if anaerobic conditions created by formation of CO2 affects the growth of the pathogen or not.

Line 293: E. faecalis and E. faecium should be in italic.

Line 301 & 302: L. monocytogenes should be in italic.

Line 306 & 316: L. monocytogenes should be in italic.

Line 332: L. monocytogenes should be in italic – please check it throughout the manuscript.

Line 351: “sakei” not “Sakei

Line 352-353: “2 log10 units” instead of “2 logarithmic units”

Line 389: You have already said that (in line 358) sucuk is a Turkish fermented sausage, no need to repeat.

Lines 398-399: Please rephrase the sentence. “…limited….below the detection limit” sounds weird.

Line 426: “storage”

Lines 426, 427, 429 & 430: all italic.

Lines 440, 441, 442, 443, 444, 445 & 446: all italic.

Lines 446, 452, 453,454 & 455: all italic – please check it throughout the manuscript.

Line 458: “frequently”

Line 461: “nisin” – please also see lines 467 and 468.

Line 490: “3 log10 units” instead of “3 logarithmic cycles” – please also see line 509.

References – Please check this section – see for example lines 654 and 890.

Author Response

We would like to thanks to the Reviewer 2 because of all the comments have been very helpful to improve quality of the paper. 
